# I CrossFit; Do You? Cross-Sectional Peer Similarity of Physical Activity Behavior in a Group High Intensity Functional Training Setting

**DOI:** 10.3390/ijerph19094932

**Published:** 2022-04-19

**Authors:** Tyler Prochnow, Christina Amo, Megan S. Patterson, Katie M. Heinrich

**Affiliations:** 1Department of Health & Kinesiology, College of Education and Human Development, Texas A&M University; College Station, TX 77843, USA; christinaamo@tamu.edu (C.A.); megpatterson@tamu.edu (M.S.P.); 2Department of Kinesiology, College of Health and Human Sciences, Kansas State University, Manhattan, KS 66506, USA; kmhphd@ksu.edu

**Keywords:** social influence, network analysis, group fitness, high intensity functional fitness

## Abstract

Physical activity (PA) is essential for physical, mental, and emotional health; however, few adults engage in enough PA. Group exercise environments such as CrossFit can promote sustained exercise habits through social influence, support, and norms. This cross-sectional study aims to provide evidence for PA social influence at CrossFit. CrossFit members (*n* = 62) reported PA, workout logging frequency, and members at their gym they: (1) work out with and (2) go to with personal matters. Separate linear network autocorrelation models (LNAMs) determined if individuals reported similar PA scores as those of their social ties at CrossFit that they work out with and/or those they go to for personal matters. Participants reported a mean of 2740.55 MET minutes/week (SD = 1847.08), working out with a mean of 9.89 members (SD = 6.26), and speaking to a mean of 2.66 members about personal matters (SD = 3.68). A person’s PA was significantly associated with that of their ties they go to with personal matters (PE_p_ = 0.08, SE_p_ = 0.02), but was not associated with the PA of their ties they work out with (PE_w_ = 0.02, SE_w_ = 0.01). Social influence on PA levels was present when a deeper connection is made between members. Fostering and promoting deeper connections between members may help promote PA and continued exercise habits.

## 1. Introduction

Physical activity (PA) has numerous preventive health benefits including reduced risk of heart disease, diabetes, obesity, hypertension, and some cancers [1]. PA is also correlated with decreased depression and anxiety, enhanced mental and emotional states, and overall higher quality of life [2]. Further, sustained and habitual exercise participation contributes to enhanced brain health, weight management, ability for everyday activities, general metabolic function, and sleep quality [1]. Currently, adults are recommended to be physically active at a moderate intensity for 150 min (or 75 min of vigorous intensity activity) per week and participate in resistance training two days per week to maintain good health [3]. Unfortunately, most American adults do not achieve the recommended amount of PA. Fifty-four percent of adults meet the minimum recommendation for aerobic activity only, 25% meet the recommendation for muscular strengthening only, and only 24% of adults meet national guidelines for both muscle strengthening and aerobic PA [4]. This deficit costs the United States USD 117 billion per year in healthcare costs associated with inadequate PA [5]. Further, recent research indicates that many adults were significantly less physically active in recent years as a result of the COVID-19 pandemic response and related precautions [6,7].

While rates of obesity and inactivity continue to rise, group exercise has also been growing in popularity [8]. Individuals involved in a fitness community show a greater adherence to PA, possibly due to the reinforcing effects of social support, influence, and norms [9]. When included in a fitness community, exercise intimidation decreases, participation enjoyment and support increase, and a sense of accomplishment is enhanced [10]. Individuals involved in a fitness community also report an increase in social cohesion and interpersonal communication skills, reduction in anxiety [11], as well as improved knowledge and attitudes towards physical exercise [2]. Including an enjoyable social component into a potentially unenjoyable or uncomfortable activity increases cognitive engagement and provides opportunities for repeated motivation to participate [12]. These elements of a positive group exercise environment also map onto constructs from social ecological frameworks theorized to improve PA behavior through several areas of influence including intrapersonal and interpersonal influences [13]. The interpersonal influences in such settings are strong indicators of social support. Further, constructs of the Social Cognitive Theory and Self-Regulation Theory, such as receiving social and physical corrective feedback as well as goal setting and tracking, are also strong components of group exercise [14].

One way to achieve the health benefits of PA in less time with greater enjoyment is high intensity interval training (HIIT) and high intensity functional training (HIFT) [15]. HIIT and HIFT have been shown to increase exercise adherence compared to other exercise alternatives such as “split routines” (i.e., alternating between aerobic and resistance training days) which require a significantly greater time commitment [15]. The main difference between HIIT and HIFT is that HIFT incorporates constantly varied exercises that elicit a greater neural response and muscle recruitment [16] for more advantageous outcomes in strength, metabolic conditioning performance, and body composition [17]. HIFT is specifically beneficial for habitual exercise considering participants’ enjoyment and connectedness are positively correlated with length of participation [18] which may negate the current 50% drop-out rate within the first six months of PA initiation [19].

A fast-growing, HIFT-style exercise that incorporates a group setting is CrossFit. CrossFit is recognized by its intensity and group based activities combining classic strength training with conditioning, gymnastics, Olympic weightlifting, and other functional movements [20]. CrossFit aims to improve cardiovascular fitness, body composition, stamina, flexibility, strength, power, balance, and anaerobic capacity [21]. The typical CrossFit participant reports higher levels of intrinsic motivations such as enjoyment, challenge, and affiliation [22] which may encourage greater adherence compared to extrinsic motives (i.e., body appearance or tone) [23]. Further, many CrossFit participants also closely track their workouts, a form of self-regulation which can improve exercise adherence [14,24]. Additionally, research suggests the community atmosphere within CrossFit is important to a participant experiencing health benefits including sustained exercise habits [25].

CrossFit is a novel modality that uses social and community engagement to motivate and inspire participants to achieve high intensities repeated for each session. Social support and positive social interactions play an important role in the PA behaviors of adults [26]. Studies have specifically documented associations between the social environment at CrossFit and higher exercise self-efficacy [27], preference for and tolerance of exercise [28], and overall participant satisfaction at CrossFit [29]. Moreover, while several studies suggest PA is influenced by social ties and social networks [30], less is known specifically about how the social structure of CrossFit may influence or be influenced by PA behaviors.

### Purpose and Hypotheses

This article specifically aims to better understand how PA behaviors are distributed across a network of CrossFit participants. Better understanding of this distribution will provide evidence for social influence within CrossFit and potentially implicate intervention strategies to improve the PA of participants. Further, it is hypothesized that PA behaviors will be clustered within the network of CrossFit participants. Stated differently, CrossFit members who are socially connected to one another (operationalized by who they would go to with a personal matter and with whom they work out) will be more similar in PA behaviors as opposed to others in their CrossFit network at large. As a sub-aim, this study will also assess whether participants who track their workouts are more physically active.

## 2. Materials and Methods

### 2.1. Participants and Procedure

This cross-sectional study involved a self-reported survey from June–September 2021. Active members over at a CrossFit gym located in Texas participated in this study. Members were eligible to participate if they were over the age of 18 and had an active membership. Prior to data collection, researchers collaborated with gym owners to identify all members at their gym who were actively participating in CrossFit classes (as opposed to personal training) at the time of data collection. As a result, gym owners provided a roster of members who would be invited to participate in the study. At the time of data collection, 102 members were invited to participate in the study and 62 completed surveys (60.8% response rate). Each person received emails from their gym owner describing the study purpose and an invitation to participate in an online survey. After giving their electronic informed consent, respondents answered questions measuring demographic information, PA and associated behaviors, and social ties present among gym members. Members could withdraw from the study at any time and the answers were kept confidential. All study procedures were approved by the institutional review board prior to data collection, and all data are available upon request.

This study incorporated a whole network (i.e., sociocentric) design to assess social ties present within a defined group at a CrossFit gym. As such, whole network research does not utilize sampling techniques, but requires enough members of a defined network provide data for accurate analysis of the defined network [31]. Research suggests a 60% response rate from the network yields an adequate enough representation of network members and their respective social ties to appropriately conduct analyses [32]. In this case, the response rate was large enough for subsequent whole network analyses.

### 2.2. Measures

Participants were asked to self-report age and gender as well as whether or not they logged or tracked their workouts with response options of “always”, “most of the time”, “about half the time”, “sometimes”, and “never” coded 4 to 0, respectively.

#### 2.2.1. Physical Activity

PA in this study was self-reported using the International PA Questionnaire—Short Form (IPAQ-SF) which is considered a valid and reliable method of self-reporting PA for adults [33]. The IPAQ-SF asks participants to report the number of days in the last week in which they participated in vigorous PA, moderate PA, and walking, separately [33]. Participants also report how many minutes they typically spent in each domain per day [33]. These inputs are then used to estimate a person’s PA in metabolic equivalent of task (MET) minutes per week [33]. MET-min/week was used as the dependent variable for each model. For descriptive purposes, the IPAQ-SF scores are categorized into three categories: high, moderate, and low [33]. To be categorized as high, an individual would have to report vigorous intensity activity on at least 3 days, achieving a minimum total PA of at least 1500 MET minutes a week or 7 or more days of any combination of walking, moderate intensity, or vigorous intensity activities achieving a minimum total PA of at least 3000 MET minutes a week [33]. To be categorized as moderate, and individual would have to report 3 or more days of vigorous intensity activity and/or walking of at least 30 min per day, 5 or more days of moderate intensity activity and/or walking of at least 30 min per day, or 5 or more days of any combination of walking, moderate intensity, or vigorous intensity activities achieving a minimum total PA of at least 600 MET minutes a week [33]. Individuals who do not meet these criteria are categorized as low [33].

#### 2.2.2. Social Connections

Social connections were operationalized in two ways within this study: (1) by asking each participant to report anyone they worked out with at the gym (“worked out with”); and (2) by asking each participant to identify any and all members of their CrossFit gym whom they would go to with a personal matter (“personal matters”). Each respondent reviewed a roster of all members at their gym and selected anyone that fit those criteria. This network generator (i.e., a question used to elicit a list of individuals important to the person being surveyed) was adapted from similar surveys [34]; however, no test–retest reliability or validity tests were conducted. Connections in the worked out with network were operationalized as undirected connections (i.e., the existence of a tie between two people was deemed reciprocal); however, connections in the personal matters network were directed connections from the participant to the person they nominated (i.e., if person A went to Person B with a personal matter, it was not assumed Person B also went to Person A with a personal matter). Duration of connections or kinship was not established.

### 2.3. Data Analysis

Linear network autocorrelation modeling (LNAM) procedures were used to model the association between one’s PA and the PA of one’s social connections while controlling for other covariates [35]. LNAM is similar to ordinary least squares regression modeling; however, it takes into account the similarity observed in the dependent variable across dyads. LNAM specifically deals with the interdependent nature of network measures and determines the role network influences and connections may play in explaining a specific outcome variable [35,36,37,38]. Separate models were calculated for each network (i.e., worked out with and personal matters). LNAMs return parameter estimates (PEs), standard errors (SEs), and associated *p*-values. The PEs for covariate terms can be interpreted as unstandardized effects. Subsequently, the network effect parameter in this case can be interpreted as a standardized effect of association between the PA of the individual and that of their connections. Descriptive statistics such as frequencies, means, and standard deviations were calculated in SPSS v.27 (IBM, Armonk, NY, USA) [39]. Network data were cleaned and managed using RStudio and the statnet package [40].

## 3. Results

Participants (*n* = 62) in this sample were a mean of 34.58 years old (SD = 9.78) with 71% (*n* = 44) identifying as female and 93.5% White. Most participants reported logging their workouts always (25.8%, *n* = 16) or most of the time (24.2%, *n* = 15); however, nearly a third reported only logging workouts sometimes (30.6%, *n* = 19). This sample was highly active and reported a mean of 2740.55 MET minutes/week (SD = 1847.08). As calculated by the IPAQ-SF, 72.6% (*n* = 45) were classified as high activity, 21.0% (*n* = 13) as moderate activity, and 6.5% (*n* = 4) as low activity [33]. Participants reported working out with a mean of 9.89 other members (SD = 6.26) but only speaking to a mean of 2.66 other members about personal matters (SD = 3.68).

Linear network autocorrelation model results indicated significant models associated with MET minutes per week for both the worked out with (R^2^ = 0.22) and personal matters (R^2^ = 0.38) networks. Full results are presented in Table 1. Age (PE_w_ = 49.99, SE_w_ = 15.24; PE_p_ = 47.01, SE_p_ = 12.70) and frequency of logging workouts (PE_w_ = 404.62, SE_w_ = 161.84; PE_p_ = 314.58, SE_p_ = 140.52) were significantly and positively associated with reported PA in both network configurations. When considering network effects, the PA of those whom an individual worked out with was not significantly associated with their own PA (PE_w_ = 0.02, SE_w_ = 0.01); however, the PA of those with whom they discussed personal matters was associated with individual level PA (PE_p_ = 0.08, SE_p_ = 0.02). Figure 1 displays these networks including their number of connections and self-reported PA. In this figure, nodes (dots) indicate each person sampled which are colored and sized according to the number of connections they reported and darker circles correspond to individuals who reported more PA. Lines between nodes indicate the connections within the network.

## 4. Discussion

CrossFit is often lauded for its ability to promote PA, improve exercise adherence, and enhance health outcomes while creating a community atmosphere. This article specifically aimed to better understand how PA behaviors are distributed across a network of CrossFit participants. Further, results here indicate CrossFit members who were socially connected to one another were more similar in PA behaviors as opposed to others in their CrossFit network at large. Specifically, social connections with whom members discussed personal matters exhibited significant similarity while those with whom they worked out with did not. Additionally, participants who tracked their workouts more frequently reported significantly more PA.

As hypothesized, the PA of those whom an individual was connected to was associated with individual level PA. However, it was noteworthy that this network effect on PA was only significant when considering the personal matters discussant network and not simply those with whom they worked out. In this manner, a deeper level of connection exhibited network similarity in PA, suggesting that connections which go beyond working out together may be needed to influence PA behavior. From the perspective of the Social Cognitive Theory and Self-Regulation Theory [14], it could mean that modeling of PA behaviors in a group of adults requires more than just observation but also trust and deeper connection. Additionally, while network theory alludes to the importance of weaker ties in the transfer or information and intel [41], it also posits behavior modification and influence are more likely to happen via stronger social ties [42].

This influence of close personal connections has been reported in past literature. Specifically, in a study investigating social networks and risk of chronic disease in the general population, individuals with more close-knit relationships reported higher levels of PA and lower levels of sedentary behavior [30]. Similarly, in a person-centered (i.e., egocentric) network study, individuals were significantly more likely to engage in regular exercise if their social connections also exercised regularly; nevertheless, this probability significantly increased if their relationship with the person was deemed close or strong [43]. Strengthening the relationships between CrossFit members beyond working out with one another to connections which can provide enhanced social support may be the key to social influence and the promotion of PA through these networks.

In this sample, the frequency of logging workouts was significantly associated with PA. This result was not surprising as logging workouts is a common and simple way to reflect on one’s achievement, while also tracking progress, serving as a constant reminder of one’s PA goals and inspiration [19]. Self-Regulation Theory and Social Cognitive Theory posit that logging helps monitor results and effectiveness for those who struggle with consistent PA [44]. Logging workouts provides an opportunity for the participant to reflect on and engage in every session with more intention [45]. Over time, logging may enhance one’s enjoyment and sense of competence toward PA when comparing historical progress [46]. Additionally, logging a workout more frequently may have social implications if the logging method involves sharing the information with others in the gym or in their personal network. This signaling of PA to others may influence those individuals who can see this log, hence providing further social influence.

Lastly, it should be noted that age was also a significant positive factor in both models. In other words, older members in the network reported significantly more PA. While PA typically decreases as individuals age [47], this sample is highly specific as they represent a highly active group participating in CrossFit. It is possible that this participation in CrossFit enhanced their PA participation even for older participants in this sample.

### 4.1. Implications for Practice

This study builds upon previous work suggesting the importance of social connections created within CrossFit [27]. As such, fostering deep and meaningful relationships between participants in CrossFit classes could result in more active members, in addition to other health benefits associated with social belonging and connection. Internally, this could translate to coaches and leaders within the CrossFit network creating opportunities to bond with other members outside of workouts. Recent research shows that a sense of community can foster greater adherence to and enjoyment in HIFT [48,49], and this study supports the effort to connect people who are active with one another. Similarly, from a network perspective, it could be advantageous to use opinion leaders and centrally positioned CrossFit members to engage and promote CrossFit with less connected members [50].

Further, this study reinforces the benefit of group-based activity options, and the role social connection can play in enhancing someone’s health. The similarity exhibited between members who are connected on a deeper level than simply working out at the same time implicates the need for enhanced social connection and social support in order for social influence to occur. Practitioners interested in promoting activity could consider the utility of suggesting and providing group-based exercise opportunities that could result in greater social connection and increased activity levels [51].

### 4.2. Limitations

There are several limitations to this study to consider. First, the cross-sectional nature of this study does not allow for the differentiation between selection and influence. Next, while the study did use previously validated self-report measures of PA, there is always a level of bias and error to consider with self-report as compared to objectively measured PA. Some studies have recommended using other measures to enhance the reliability and validity of the IPAQ [52]. More rigorous measurements of PA would substantially improve these findings. This study represents one CrossFit gym and a limited number of participants. Future research may wish to include varying networks to investigate these phenomena on a broader scale and across different groups. Further, while the social connections measure has been used in past studies and is based on prior work, there was no reliability or validity measurements conducted. Similarly, duration of connection or possible kinship between members may have additional implications and may have added bias. Additionally, this sample was highly specific, including only CrossFit participants and not the general population which may not exhibit similar characteristics. Lastly, the sample was also largely skewed towards White women which may limit the generalizability.

## 5. Conclusions

Despite these limitations, this study provides evidence that the social structure present within CrossFit gyms is associated with PA behaviors. Specifically, deeper connections, not just working out with other members, may be driving the similarity seen within these networks. Further understanding and fostering these connections may be the key to promoting PA through social influence in these programs.

## Figures and Tables

**Figure 1 ijerph-19-04932-f001:**
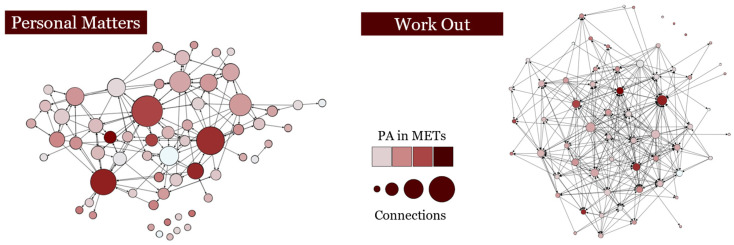
Personal matters and work out networks displayed. Note: nodes are sized by the number of connections in each network and colored by the amount of reported physical activity.

**Table 1 ijerph-19-04932-t001:** Linear network autocorrelation models for MET minutes per week.

	Work Out Network			Personal Matters Network		
Parameters	R^2^ = 0.22			R^2^ = 0.38		
	Estimate	Std. Error	*p*-Value	Estimate	Std. Error	*p*-Value
**Age**	49.99	15.24	<0.01 *	47.01	12.70	<0.01 *
**Gender**	−749.03	483.75	0.12	−557.42	417.65	0.18
**Logging Workout**	404.62	161.84	0.01 *	314.58	140.52	0.03 *
**Network Effects**	0.02	0.01	0.21	0.08	0.02	<0.01 *

* Indicates a significant effect at *p* < 0.05.

## Data Availability

Data can be made available upon reasonable request.

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
