# Peer review of "I CrossFit; Do You? Cross-Sectional Peer Similarity of Physical Activity Behavior in a Group High Intensity Functional Training Setting"

_ijerph, 2022, doi:10.3390/ijerph19094932_

Round 1
Reviewer 1 Report
The work presents results related to the practice of CrossFit, a modality of the fitness universe that is growing. Therefore, it may be interesting for some specialized readers to know more about this modality. However, I leave here some comments directed to the methodological nature of the article:
- Please, describe the study’s design with Title and/or abstract; I really like the title, but we should know what the work study design is.
- This approach that relates PA and social influence is interesting, but it can confuse the reader, maybe. Talking about PA in an article focused on a specific type of physical exercise can also cause confusion. There is a paradigm that talks about systematic physical exercise increasing or not physical activity levels. I think they can try to establish a theoretical causal relationship there. This idea could also generate an increase in statistical treatment and results.
- In the introduction section please, included before the state of specific objectives, the prespecified study hypotheses
- In the method section (I recommend that you create other subsections to respond to the following recommendations): i) please, describe the study design (i.e cross-sectional, experimental ???) ii) present key elements of study design early in the paper such as Describe the setting, locations, and relevant dates, including periods of recruitment, exposure, follow-up, and data collection, etc; iii) give the eligibility criteria, and the sources and methods of selection of participants;
- Please, describe any efforts to address potential sources of bias
- Regarding participants: i) report numbers of individuals at each stage of study—eg numbers potentially eligible,examined for eligibility, confirmed eligible, included in the study, completing follow-up, and analysed.
- Participants: Give general characteristics. Please present a table with this information;
- Statistical analysis: the option for the correlation model is very interesting, however, it will be necessary to explain the figure better. For this study, no additional analysis was performed ? If yes, please report.
Author Response
I CrossFit; Do you? Cross-sectional peer similarity of physical activity behavior in a group high intensity functional training setting
|
Reviewer Comments |
Authors’ response |
Location in document |
|
Reviewer 1 |
||
|
Please, describe the study’s design with Title and/or abstract; I really like the title, but we should know what the work study design is. |
||
|
This approach that relates PA and social influence is interesting, but it can confuse the reader, maybe. Talking about PA in an article focused on a specific type of physical exercise can also cause confusion. There is a paradigm that talks about systematic physical exercise increasing or not physical activity levels. I think they can try to establish a theoretical causal relationship there. This idea could also generate an increase in statistical treatment and results. |
||
|
In the introduction section please, included before the state of specific objectives, the prespecified study hypotheses |
||
|
In the method section (I recommend that you create other subsections to respond to the following recommendations): i) please, describe the study design (i.e cross-sectional, experimental ???) ii) present key elements of study design early in the paper such as Describe the setting, locations, and relevant dates, including periods of recruitment, exposure, follow-up, and data collection, etc; iii) give the eligibility criteria, and the sources and methods of selection of participants |
||
|
Please, describe any efforts to address potential sources of bias |
||
|
Regarding participants: i) report numbers of individuals at each stage of study—eg numbers potentially eligible,examined for eligibility, confirmed eligible, included in the study, completing follow-up, and analysed. |
||
|
Participants: Give general characteristics. Please present a table with this information; |
||
|
Statistical analysis: the option for the correlation model is very interesting, however, it will be necessary to explain the figure better. For this study, no additional analysis was performed ? If yes, please report. |

Reviewer 2 Report
Interesting title, so I am happy to assist with reviewing this paper.
Abstract is well written and informative, but key words, when looking at the title, lack notification about intensity factor.
Introduction is basic. I would expect some more introduction into the rationale for the study. For example, obesity that you mention in the Introduction is a global issue, so you could show some trends in epidemiological obesity among adults in various countries and provide some reasons, especially bearing in mind the last 2-3 years of pandemic lockdown and comparison of physical activity before and during the lockdown pandemic.
Another issue is the problem of intensity. You are right that it is possible to gain better health effects with HITT, but enjoyment of participants may depend on the level of intensity. For example look into more on how intensity levels are related with undertaking PA ('Understanding the Motives of Undertaking Physical Activity with Different Levels of Intensity among Adolescents: Results of the INDARES Study'), especially that you mention the atmosphere in Crossfit as an important mediating factor. And this may also differ according to cultural and social backgrounds and even gender (look for example into: Psychosocial Determinants of Participation in Moderate-to-Vigorous Physical Activity among Hispanic and Latina Middle school-aged girls.
The idea of social interaction among peers as a mediating factor connected to Crossfit participation seems interesting, but here again, I would expect you to expand a little bit on the role of support in undertaking PA/Crossfit.
Generally, Introduction without theoretical framework needs some strengthening of the rationale of the study.
Methods
In the description of the participants and selection procedure I think it would be worth mentioning whether participants could withdraw from the research at any time, and whether their answers were anonymous.
Research tools have been described sufficiently, and although I am not very keen of IPAQ (as there have been some issue reported in various studies - perhaps you could also look into those and mention it in limitations - look into 'Validity of the International Physical Activity Questionnaire (IPAQ) for assessing moderate-to-vigorous physical activity and sedentary behaviour of older adults in the United Kingdom or Progress and pitfalls in the use of the International Physical Activity Questionnaire (IPAQ) for adult physical activity surveillance - just to show your critical approach to testing such small samples with IPAQ). I know that you mention it in Limitation section, but be more critical about it.
Also, Social connection tool - how was it developed? Did you have any procedure for that? Has it been tested for reliability? Did you check for family connections between the Cross-fit community? This could also be an problematic matter in terms of social interactions (husband-wife, sister-sister, other members of extended family ect.). I would expect some more information on 'personal matters' - what did you mean by that term, and whether it was clear for the respondents?
Statistical methods sophisticated but in accordance with the research line.
In Results I find quite interesting illustration of the results on figure 1.
In Discussion you mention Self-Regulation Theory ... I think it would be better if this theory be introduced earlier on in the paper (like in Introduction section). In the Introduction section you mention 'enjoyment' as one of the factors influencing PA, whereas there is nothing about it in Results and you come back to it in Discussion. I believe referring to support as a valid source of motivation towards leisure PA should be more accentuated in this section, as research evidence suggests strong associations between adolescents' physical activity behavior and their perception of peer and even teacher support, which later may be reflected in their expectation in a free environment of leisure PA of their own choice.
Tables and figures are neat and can be read easily.
References - this section needs to be extended accordingly to inclusion of more works in the body of the text.
Author Response
|
Abstract is well written and informative, but key words, when looking at the title, lack notification about intensity factor. |
||
|
Introduction is basic. I would expect some more introduction into the rationale for the study. For example, obesity that you mention in the Introduction is a global issue, so you could show some trends in epidemiological obesity among adults in various countries and provide some reasons, especially bearing in mind the last 2-3 years of pandemic lockdown and comparison of physical activity before and during the lockdown pandemic. |
||
|
Another issue is the problem of intensity. You are right that it is possible to gain better health effects with HITT, but enjoyment of participants may depend on the level of intensity. For example look into more on how intensity levels are related with undertaking PA ('Understanding the Motives of Undertaking Physical Activity with Different Levels of Intensity among Adolescents: Results of the INDARES Study'), especially that you mention the atmosphere in Crossfit as an important mediating factor. And this may also differ according to cultural and social backgrounds and even gender (look for example into: Psychosocial Determinants of Participation in Moderate-to-Vigorous Physical Activity among Hispanic and Latina Middle school-aged girls. |
||
|
The idea of social interaction among peers as a mediating factor connected to Crossfit participation seems interesting, but here again, I would expect you to expand a little bit on the role of support in undertaking PA/Crossfit. |
||
|
Generally, Introduction without theoretical framework needs some strengthening of the rationale of the study |
||
|
In the description of the participants and selection procedure I think it would be worth mentioning whether participants could withdraw from the research at any time, and whether their answers were anonymous. |
||
|
Research tools have been described sufficiently, and although I am not very keen of IPAQ (as there have been some issue reported in various studies - perhaps you could also look into those and mention it in limitations - look into 'Validity of the International Physical Activity Questionnaire (IPAQ) for assessing moderate-to-vigorous physical activity and sedentary behaviour of older adults in the United Kingdom or Progress and pitfalls in the use of the International Physical Activity Questionnaire (IPAQ) for adult physical activity surveillance - just to show your critical approach to testing such small samples with IPAQ). I know that you mention it in Limitation section, but be more critical about it. |
||
|
Also, Social connection tool - how was it developed? Did you have any procedure for that? Has it been tested for reliability? Did you check for family connections between the Cross-fit community? This could also be an problematic matter in terms of social interactions (husband-wife, sister-sister, other members of extended family ect.). I would expect some more information on 'personal matters' - what did you mean by that term, and whether it was clear for the respondents? |
||
|
In Discussion you mention Self-Regulation Theory ... I think it would be better if this theory be introduced earlier on in the paper (like in Introduction section). In the Introduction section you mention 'enjoyment' as one of the factors influencing PA, whereas there is nothing about it in Results and you come back to it in Discussion. I believe referring to support as a valid source of motivation towards leisure PA should be more accentuated in this section, as research evidence suggests strong associations between adolescents' physical activity behavior and their perception of peer and even teacher support, which later may be reflected in their expectation in a free environment of leisure PA of their own choice. |

Round 2
Reviewer 2 Report
Text has been enhanced as some suggestions have been taken into account, thought still I have some doubts, but let it go., I don't think you are able to enhance it any further.